# Kaolin-Enhanced Superabsorbent Composites: Synthesis, Characterization and Swelling Behaviors

**DOI:** 10.3390/polym13081204

**Published:** 2021-04-08

**Authors:** Mengna Chen, Xuelong Chen, Caiyan Zhang, Baozheng Cui, Zewen Li, Dongyu Zhao, Zhe Wang

**Affiliations:** 1Department of Polymer Materials and Engineering, School of Chemistry and Materials Science, Heilongjiang University, Harbin 150080, China; Mengna999456@163.com (M.C.); zhangcaiyan061995@163.com (C.Z.); 2201459@s.hlju.edu.com (B.C.); lizewen145@126.com (Z.L.); zhaodyu@aliyun.com (D.Z.); 2School of Material Science and Engineering, Nanyang Technological University, 50 Nanyang Avenue, Singapore 639798, Singapore; xchen014@e.ntu.edu.sg

**Keywords:** sodium alginate, starch, kaolin, free radical polymerization, superabsorbent composites, characterization, swelling behavior

## Abstract

One type of low-cost and eco-friendly organic‒inorganic superabsorbent composite (SAPC) was synthesized by free radical polymerization of acrylic acid (AA), starch (ST), sodium alginate (SA) and kaolin (KL) in aqueous solution. The structure and morphology of the SAPC were characterized by Fourier transform infrared spectrometer (FT-IR), scanning electron microscope (SEM), X-ray diffraction (XRD) and thermogravimetric analysis (TGA). The influence of different reaction conditions on water absorption of SAPC, i.e., SA and KL contents, AA neutralization degree (ND), potassium persulfate (KPS) and N, N′-methylenebisacrylamide (MBA) loading were systematically studied. Under the optimal synthesis conditions, very high water absorption of 1200 g/g was achieved. The swelling kinetic mechanism of SAPC was studied by pseudo-second order swelling kinetics model and Ritger‒Peppas model. The performances of SAPC under different environments were tested and results revealed that this new SAPC had excellent swelling capacity, high water retention, good salt tolerance in monovalent salt solution (NaCl solution) and good pH tolerance between 4 and 10.

## 1. Introduction

Superabsorbent polymers (SAPs) are one type of three-dimensional (3D), cross-linked hydrophilic materials that can absorb and hold a large amount of water within a certain period of time [1,2,3,4]. Before the emergence of SAP, natural biomass materials, such as cotton, have been used for traditional absorbents [5]. These materials absorb water through capillarity, yet with low water absorption capacity, and cannot retain water for a long time, making them unsuitable for many life and industry applications. By contrast, SAP mainly absorbs water through chemical adsorption, with efficient water absorption and large water retention capacity [6]. Owing to excellent performance, SAP has been widely used in diversified fields, such as personal care, industry, chemicals, forestry, agriculture, wastewater treatment and drug-delivery systems [7,8,9]. 

Although with wide applications, SAP still has several key issues that need to be addressed, such as high synthesis costs, potential toxicity and serious environmental influences [10,11]. To reduce production costs and improve performance, many inorganic clay minerals (montmorillonite, kaolin, bentonite and attapulgite) and natural polymeric materials such as polysaccharides (starch, sodium alginate, cellulose, chitosan and their derivatives) have been used to synthesize environmentally friendly organic‒inorganic superabsorbent composites (SAPCs) [12,13,14]. 

Kaolin is a hydrous aluminosilicate with layered structure, which has advantages of large specific surface area, large cation exchange capacity and low cost [15,16]. The addition of kaolin can not only significantly reduce the production cost, but is also able to improve water absorption, gel strength, thermal stability and mechanical properties of the final products [17,18]. 

Starch is a natural polysaccharide, mainly composed of amylose and amylopectin [19]. The former possesses a linear chain composed of α-1,4-linked glucose unites, while the latter has a highly branched chain of α-1,4-linked glucose units interlinked by α-1,6-linked bonds [20]. Sodium alginate, an abundant natural polysaccharide extracted from various species of brown seaweed, is composed of β-1,4-linked D-mannuronic acid and α-1,4-linked L-guluronic acid with various proportions, which can be arranged into different sequences in the polymeric backbone [21,22]. Taking advantages of excellent properties, such as biocompatibility, biodegradability, low cost and low toxicity, starch and sodium alginate have become ideal backbones for the synthesis of SAP [23]. By now, graft polymerization of vinyl monomers such as acrylamide, acrylonitrile, and acrylic acid with starch and sodium alginate, respectively, has been reported. Erdener Karadağ et al. synthesized novel composite sorbent AAm/MA hydrogels containing starch and kaolin used for water sorption and dye uptake [24]. Jihuai Wu et al. fabricated one type of starch-graft-acrylamide/kaolin superabsorbent composite with water absorbency 4000 times higher than its own weight [25]. Linhui Zhu et al. studied the adsorption behaviors of sodium alginate graft poly(acrylic acid-co-2-acrylamide-2-methyl propane sulfonic acid)/kaolin hydrogel composite towards dyes [26]. Sodium alginate graft poly(acrylic acid-co-acrylamide)/kaolin composite hydrogel has also been prepared by Yaoji Tang et al [27]. However, there are few reports on the simultaneous introduction of starch and sodium alginate into the system to graft polymerize with vinyl monomer. It is reported that the use of a polysaccharide mixture in SAP synthesis shows better mechanical stability and water retention capacity compared to a single polysaccharide [28]. The addition of sodium alginate can improve pH sensitivity of starch-based SAP [29]. In addition, the preparation route of SAP from sodium alginate nowadays is attractive as to make full use of marine resources and reduce manufacture cost and pollution. Nevertheless, the viscosity of sodium alginate solution is extremely high. During the synthesis of alginate-based SAP, the viscosity of the reaction system often sharply increases due to the addition of sodium alginate, which restricts the movement of reactants and reduces the monomer conversion rate. It is hypothesized that adding a certain amount of starch can reduce the content of sodium alginate and viscosity of the system.

Based on these reasons, a novel organic‒inorganic SAPC was synthesized by free radical copolymerization with acrylic acid, starch, sodium alginate and kaolin. The structure of the SAPC was characterized by Fourier transform infrared spectrometer, scanning electron microscope, X-ray diffraction and thermogravimetric analysis. The influence of different reaction conditions on water absorption of SAPC was studied. A pseudo-second order kinetics model and Ritger‒Peppas model also illustrated the swelling kinetic mechanism of SAPC. The performance of this new SAPC was tested. Results showed that the SAPC had excellent swelling capacity, salt tolerance, water retention and pH sensitivity.

## 2. Experimental

### 2.1. Materials

Starch (ST, food grade) was purchased from Shandong Lejiake Food Co., Ltd. (Dezhou, Shandong, China). Sodium alginate (SA, analytical grade) was from Shanghai Yuanye Biotechnology Co., Ltd. (Shanghai, China). Acrylic acid (AA, analytical grade) was purchased from Tianjin Zhiyuan Chemical Reagent Co., Ltd. (Tianjin, China), kaolin (KL, chemically pure) was purchased from Shanghai Fengxian Fengcheng Reagent Factory (Shanghai, China), and potassium persulfate (KPS, analytical grade) was purchased from, Tianjin Komiou Chemical Reagent Co., Ltd. (Tianjin, China). N, N′-methylenebisacrylamide (MBA, chemically pure) was obtained from Tianjin Guangfu Research Institute of Fine Chemical (Tianjin, China). Sodium hydroxide (NaOH, analytical grade) was supplied by Tianjin Tianli Chemical Reagent Co., Ltd. (Tianjin, China). All solutions were prepared with distilled water.

### 2.2. Synthesis of SAPC

The ST (3 g) was dissolved in distilled water (40 mL) and then placed in a 500 mL four-port flask with a condenser, a nitrogen tube and a mechanical stirrer. The solution was heated to 85 °C and continuously stirred for 1 h. Then, the solution of SA was added to the flask and continued to stir until temperature dropped to 50 °C, and then KPS was added to generate radicals. After the temperature reached 70 °C and stirring for 15 min, a mixture containing KL, MBA and partially neutralized AA (20 mL AA, neutralized AA with 25% (*w*/*w*) NaOH solution under ice bath conditions) was added to the above system, followed by reaction for 3 h. A nitrogen purge was used throughout the process. The product was dried in a vacuum oven at 60 °C until the weight stabilized. Afterwards, the product was milled, and the size of all powder products used for testing was about 80 mesh.

For comparison, the synthesis of neat SAP was carried out in the same way except without KL. The structure of ST and SA is shown in Scheme 1, while the mechanistic pathway for synthesis of SAPC is shown in Scheme 2.

### 2.3. Characterization

A Fourier transform infrared spectrometer (FT-IR; Bruker, Equinox 55, Mannheim, Germany) was used to record the infrared spectra of raw materials and products in the range of 4000–400 cm^−1^. The morphology of the products was observed with a scanning electron microscope (SEM; Hitachi, S-4800, Tochigi, Japan). The crystal structure of the sample was examined by X-ray diffraction (XRD; Bruker, D8 ADVANCE, Mannheim, Germany). The thermal stability of the products was analyzed by thermogravimetric analysis (TGA; Mettler, TGA/DSCI, Greifensee, Switzerland) from 35 to 800 °C with heating rate 10 °C min^−1^ under nitrogen atmosphere.

### 2.4. Performance Test of SAPC

#### 2.4.1. Swelling Ratios

To determine the swelling ratio, dry composites (0.1 ± 0.001 g) were immersed in 500 mL of solutions (distilled water, various salt solutions and pH solutions) for 4 h at room temperature to achieve equilibrium. The swollen product was filtered through a 100-mesh screen to remove excess moisture and then weighed. Each specimen was tested thrice, and the average value was used. The water absorbent capacity was calculated using the following Equation (1): (1)$$Q_{eq}\;=\;\frac{{(m}_{1}{\;-\;m}_{0})}{m_{0}}$$ where *Q_eq_* is the water absorbent capacity and *m*_0_ and *m*_1_ are the mass of the dried and swollen products, respectively.

#### 2.4.2. Swelling Kinetics

The pre-weighed dry product (0.1 ± 0.001 g) was added into distilled water (500 mL), and swelled for a period of time (5, 10, 15, 30, 45, 60, 120, 180 and 240 min). The water absorption at time *t* was measured by the method described in Section 2.4.1.

#### 2.4.3. Swelling in Salt Solutions

NaCl and FeCl_3_ were used to prepare salt solutions with different concentrations (0.02–0.1 mol/L). The pre-weighed dry product (0.1 ± 0.001 g) was dispersed in various salt solutions (500 mL), and water absorption was measured by the method described in Section 2.4.1.

#### 2.4.4. Water Retention Measurement 

The fully swollen product (80 g) was placed in petri dishes at various temperatures (25, 45 and 60 °C), and then weighted every 2 h. According to Equation (2), water retention of the product was calculated:(2)$$W_{r}\;=\;\frac{{(M}_{t}{\;-\;M}_{0})}{{(M}_{1}{\;-\;M}_{0})}\;\times 100\%$$ where *W_r_* is the water retention rate of product, *M_t_* is the mass of the swollen product at time *t*, *M*_0_ is the mass of the dry product and *M*_1_ is the initial weight of the fully swollen products.

#### 2.4.5. Swelling Measurement in Buffer Solutions

The buffer solutions (hydrochloric acid and sodium hydroxide solutions) with pH values from 2 to 12 were prepared to test the pH sensitivity of the SAPC. A pH-meter (PHS-3C, accuracy is ± 0.01) provided by Shanghai INESA Scientific Instrument Co., Ltd. (Shanghai, China) was used to detect the pH value of the solution. Then, the dry product (0.1 ± 0.001 g) was dispersed in various buffer solutions (500 mL), and water absorption was calculated according to Equation (1).

## 3. Results and Discussion

### 3.1. FT-IR Analysis

The FT-IR spectra of the ST, SA, SAP, SAPC and KL are shown in Figure 1. The FT-IR spectrum of ST is shown in Figure 1a, peaks at 1158, 1081 and 1005 cm^−1^ were the stretching vibration of the C‒O‒C bond [30,31]. After polymerization, these peaks shifted and weakened. As shown in the spectra of SA (Figure 1b), the strong bands at 1610 and 1410 cm^−1^ were due to the asymmetric and symmetric stretching vibrations of ‒COO^−^ groups, respectively [32]. The peak at 1032 cm^−1^ was related to the stretching vibration of C‒OH [33], which moved to 1030 cm^−1^ and 1036 cm^−1^, respectively, after polymerization, and the intensity of this peak weakened significantly. Figure 1c shows the FT-IR spectrum of SAP, the absorption peak at 3469 cm^−1^ was attributed to the stretching vibration of ‒OH, and three new peaks appeared at 1711, 1567 and 1409 cm^−1^. The peak at 1711 cm^−1^ may be related to the ester group formed during graft polymerization. Also, the new absorption at 1567 and 1409 cm^−1^ could be ascribed to the asymmetric and symmetric stretching of ‒COO^−^, respectively. These results showed that both ST and SA were involved in the grafting reaction. In the infrared spectrum of KL (Figure 1e), the absorption peaks of ‒OH were observed at 3696‒3621 cm^−1^ [25]. As revealed in Figure 1d,e, the ‒OH peak shifted from 3469 cm^−1^ to 3457 cm^−1^, and a characteristic Si‒O peak was observed at 471 cm^−1^, which indicated that ‒OH on KL was also involved in the reaction.

### 3.2. Morphology

The appearance of SAPC at different states is exhibited in Figure 2a,b. The dried SAPC was a light-yellow solid, while the swollen SAPC was a transparent hydrogel. The volume of dried SAPC was significantly smaller than fully swollen SAPC. 

SEM of dried SAP and SAPC are shown in Figure 2c,d. The surface of SAP was compact and smooth, making it not conducive to the liquid entering into the polymer network. By contrast, after addition of KL, the surface of SAPC displayed a more rough and loose morphology with porous structure.

### 3.3. XRD Analysis

The XRD patterns of KL, SAPC and SAP are shown in Figure 3. The strong peak of KL at 2θ = 12.59° had an interplanar distance of d = 0.702 nm. This peak shifted to 2θ = 12.44° (0.711 nm) in SAPC due to the intercalation of KL by polymer network. No obvious peaks were observed for SAP due to the amorphous structure. In contrast, SAPC showed the typical crystallite reflections associated with KL, which indicated that KL was uniformly dispersed in the polymer matrix, and an amorphous superabsorbent composite was synthesized.

### 3.4. TGA Analysis

The TGA curves of SAP and SAPC (4 wt. % KL) are shown in Figure 4. In the range from 35 to 203 °C, the minor weight loss of SAP and SAPC was mainly ascribed to the evaporation of adsorbed moisture and bound water. The weight loss of SAP and SAPC at 203–343 °C were related to the dehydration of saccharide rings and breaking of the C‒O‒C bond in ST and SA chain. Also, the weight loss from 343 °C to 416 °C corresponded to the decomposition of the carboxyl groups of the copolymers. The major mass loss of SAP and SAPC occurred in the temperature range of 416‒480 °C and 416‒495 °C, which was due to the decomposition of the 3D network structure. It is noted between 343 and 495 °C that the decomposition rate of SAP was obviously higher than that of SAPC. Additionally, the weight residual of SAPC was 49.5%, 6.4% higher than that of SAP. These facts seem to indicate improved thermal stability of SAPC compared to that of SAP, probably owing to the heat dissipation impedance effect of KL.

### 3.5. Effects of Reaction Conditions on SAPC Performance

#### 3.5.1. Effect of SA Content

The effect of SA content on water absorption of SAPC is shown in Figure 5a. Water absorption increased from 736 to 933 g/g, with the SA content changing from 10 wt. % to 15 wt. %, and then decreased to 763 g/g, with SA content further increasing to 20 wt. %. When the content of SA, which act as the basic skeleton, was too low, AA self-polymerized, resulting in the inability to form an effective 3D network to absorb water. With much higher SA content, sharply increased viscosity resulted in the system, restricting the monomer movement and reducing the conversion rate. Thus, a less developed 3D polymer network and suppressed water absorption capacity could be expected.

#### 3.5.2. Effect of KL Content

Figure 5b depicts the effect of KL content on water absorption. Water absorption of SAPC first increased and then decreased with the increase of KL content from 3 wt. % to 7 wt. %. KL with a large number of hydrophilic groups could be chemically cross-linked with a polymer chain, forming a cross-linked polymer network with KL particles as the additional cross-linking points. The introduction of appropriate KL particles weakens the hydrogen-bonding interaction between carboxyl groups in the polymer and reduces the physical entanglement of the grafted polymer network chain [34,35]. Nevertheless, with excessive KL (>4 wt. %), the excess KL particles increased the degree of cross-linking, leading to a decrease of the penetration space of water molecules. In addition, the extra kaolin particles were also filled into the polymer network space in the form of physical filling. Yet the water absorption rate of KL itself was low, as a result, the water absorption of SAPC inevitably decreased. Therefore, appropriate KL loading is critical to obtaining SAPC with high water absorption capacity.

#### 3.5.3. Effect of Neutralization Degree of AA

The water absorptance capacity of SAPC was closely related to the neutralization degree (ND) of AA (Figure 5c). AA has higher reaction activity and polymerization rate than sodium acrylate [36]. Water absorption of SAPC increased as ND increased from 65% to 80%. When ND was lower, the polymerization reaction completed rapidly within a short time, resulting in a highly cross-linked network structure with low swelling capacity. With the increase of ND, the polymerization rate decreased correspondingly. In addition, the increase of ‒COO^−^ groups and Na^+^ content in the polymer network increased both the repulsive force between the anions on the polymer chains and the osmotic pressure difference between the inside and outside of the polymer network, which was conducive to the entry of water or other small molecules [37,38,39]. However, when ND was higher than 80%, water absorption dropped significantly, attributed to the reaction of ‒COO^−^ groups with excess Na^+^, resulting in weakening of the repulsive force.

#### 3.5.4. Effect of KPS Content

Water absorption was significantly affected by the initiator concentration and the average kinetic chain length. With low KPS content (0.3–0.9 wt. %), an integral polymer network could not be formed due to the few grafting points and a large number of unreacted monomers in the reaction system. With too much KPS (0.9–1.1 wt. %), the excessive free radicals terminated the propagating chains earlier, shortening the average kinetic chain length. In both scenarios, low water absorption capacity resulted (Figure 5d).

#### 3.5.5. Effect of MBA Content

The effect of the MBA content on water absorption of SAPC is shown in Figure 5e. Water absorption of SAPC increased first and then decreased with the increase of MBA content from 0.1 to 0.2 wt. %, and a maximum value of 1156 g/g was achieved with 0.15 wt. % MBA loading. Lower or higher MBA content resulted in reduced water absorption. The water absorption capacity of resin was closely related to its spatial network structure. When the content of MBA was not sufficient, cross-linking density was low and a complete spatial network structure could be established, leading to low water absorption and poor mechanical properties of SAPC. However, a higher concentration of MBA produced a denser cross-linked structure, which made it difficult for liquid molecules to enter.

The optimal reaction conditions of SAPC were when the masses of ST, SA, KL, KPS and MBA were 15 wt. %, 15 wt. %, 4 wt. %, 0.9 wt. % and 0.15 wt. % of AA, respectively. The neutralization degree of AA was 80%. In this work, SAP and SAPC were synthesized under the optimal conditions for further tests.

### 3.6. Performance Tests of SAPC

#### 3.6.1. Swelling Kinetics of SAPC in Distilled water

The pseudo-second order swelling kinetics model (Equation (3)) and the Ritger‒Peppas model (Equation (4)) were used to analyze the experimental swelling data to evaluate the swelling behavior of SAPC, with results shown in Figure 6a,b [40,41,42]:(3)$$\frac{t}{q_{t}}\;=\;\frac{1}{{(k}_{2}q_{e}^{2})}\;+\;\frac{t}{q_{e}}$$
(4)$$F\;=\;\frac{q_{t}}{q_{e}}{\;=\;\mathrm{kt}}^{n}$$

To calculate *n* and *k*, take the natural logarithm of Equation (4): (5)$$ln\;F\;=\;ln\;(\frac{q_{t}}{q_{e}})\;=\;ln\;k\;+\;nln\;t$$ where *t* is absorption time and *k_2_* is the rate constant of the pseudo-second order model. *q_t_* and *q_e_* correspond to water absorption of SAPC at time *t* and at equilibrium, respectively. *F* denotes fractional uptake at time *t*. *k* and *n* are the characteristic constants of the polymer and the diffusion index, respectively.

From Figure 6a, water absorption of SAPC increased rapidly in the first 15 min, reached swelling equilibrium at 15 min, and then stabilized after 15 min. The maximum water absorption was 1200 g/g. It was also found that the relationship curve between *t* and *t/q_t_* was linear with R^2^ value (0.99935) close to 1, indicating that the swelling behavior of SAPC could be fitted by the pseudo-second order kinetics model. Since this model is based on chemical adsorption assumptions, it suggests that the chemisorption was the main way of the water absorption process. 

Based on the polymer chain relaxation rate and the relative diffusion rate of water into the polymer network, the water diffusion mechanism can be classified into five types. That is, pseudo-Fickian diffusion (*n* < 0.5), Fickian diffusion (*n* = 0.5), non-Fickian diffusion (0.5 < *n* < 1.0), Case II transport diffusion (*n* = 1) and relaxed diffusion (*n* > 1) [43,44,45]. These equations were applied to the initial stages of swelling. According to Figure 6b, within 0–15 min, 0.5 < *n* < 1, indicating that the water diffusion mechanism was consistent with the non-Fickian diffusion mechanism, the diffusion and relaxation were considered to be isochronally effective. After 15 min, *n* < 0.5, the main reason of swelling was the diffusion of water molecules in the polymer network. 

#### 3.6.2. Swelling Behavior in Salt Solutions

The influences of cation and saline type on water absorption were studied using FeCl_3_ and NaCl, with results shown in Figure 6c. For the same saline solution, as cation concentration increased, water absorption of SAPC decreased. This result was due to the charge screening effect of cations, which reduced the osmotic pressure difference between the polymer network and the external saline solution and decreased the repulsive force between the ‒COO^−^ groups of the polymer chain. Comparing the two absorption curves, it was found that water absorption of SAPC was lower in the multivalent cation solution. This is attributed to the complexation between carboxylate anions of chains and multivalent cations, resulting in an increase of the network cross-linking density and a drastic decrease in water absorption [46,47].

#### 3.6.3. Water Retention Properties

The water retention of fully swollen SAPC at different temperatures, i.e., 25, 45 and 60 °C, was studied, and the results are shown in Figure 6d. As can be seen, water retention of SAPC decreased with time, and the water retention curve at 25 °C was gentler than that at 45 and 60 °C. At 25 °C, the water retention rate was about 80% after 12 h. However, the water retention rate of SAPC at 45 °C and 60 °C still reached about 37% and 6% after 12 h, respectively. The above results showed that SAPC had excellent water retention ability and thus broad application prospects.

#### 3.6.4. Swelling Behavior in Buffer Solutions

As a new superabsorbent composite, the swelling capacity in different pH solutions has great influence on its application in various fields. Hence, the swelling behavior of SAPC in different buffer solutions was studied, with results shown in Figure 6e and the mechanism illustrated in Figure 6f. At low pH values (pH ≤ 4), due to the higher concentration of H^+^, most of the ‒COO^−^ groups were transformed into ‒COOH groups, which weakened the electrostatic repulsion between ‒COO^−^ groups and strengthened the hydrogen-bonding interaction between the ‒COOH groups of polymer chains [48]. Therefore, water absorption was low. With an increase of pH value (4 ≤ pH ≤ 7), water absorption of SAPC increased, owing to gradual ionization of ‒COOH groups, which resulted in an increase of electrostatic repulsion between ‒COO^−^ groups. At a high pH value (pH ≥ 7), as Na^+^ concentration increased in the solution, water absorption decreased correspondingly. The decrease was due to the charge shielding effect of excessive Na^+^ on the ‒COO^−^ groups, which weakened the electrostatic repulsion force and reduced the osmotic pressure difference between the inside and outside of the polymer network. Generally, this type of SAPC had excellent pH tolerance between 4 and 10.

## 4. Conclusions

To summarize, a series of SAPC was successfully prepared by free radical polymerization of SA, ST, AA and KL in aqueous solution. The optimal reaction conditions of SAPC were achieved when the mass of ST, SA, KL, KPS and MBA were 15 wt. %, 15 wt. %, 4 wt. %, 0.9 wt. % and 0.15 wt. % of AA, respectively, with a neutralization degree of AA 80%. With optimized reactants concentrations, a maximum water absorption of 1200 g/g was obtained. FT-IR spectra confirmed the success of the polymerization reaction. SEM images revealed the rough and porous surface of SAPC which was conducive to the liquid entering into the polymer network. XRD patterns proved that KL was uniformly dispersed in the polymer matrix. TGA spectra indicated that the addition of KL improved the thermal stability of SAPC. The swelling kinetic of SAPC was studied, showing that the swelling behavior of SAPC followed the pseudo-second order kinetic model and the non-Fickian diffusion model. Along with excellent water absorption capacity, the fabricated SAPC also had excellent water retention property, good salt tolerance in monovalent salt solution and pH tolerance between 4 and 10, making it promising in many applications.

## Data Availability

Data availability upon request.

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
