# Peer review of "Kaolin-Enhanced Superabsorbent Composites: Synthesis, Characterization and Swelling Behaviors"

_polymers, 2021, doi:10.3390/polym13081204_

Round 1
Reviewer 1 Report
Are there any studies done with starch and kaolin?
What NaOH concentration was used?
Are they suggesting to the research team that starch synthesis be done with metakaolin?
There are numerous studies on the benefits of using metakaolin in the polymerization process instead of kaolin. It is proposed to do research on metakaolin and starch.
Does the organic phase from kaolin interfere with the polymerization process?
Reviewer 2 Report
The materials are presented for this manuscript by the authors are original and interesting. However, the quality of this manuscript should be improved by the following comments are listed below:
- The abstract is poorly written without containing the key information regarding the findings of results obtained.
- In the swelling test, the weight of the sample was constant (0.1 g). How was the weight controlled? I recommend recording the range.
- The results are presented in Figure 5 should discuss in detail based on the results obtained by depending on comparative values and differences of swelling tests
- The conclusions obtained represent nothing in relation to the results obtained and should be supported by data.
Author Response
Reviewer: 2
Comment 1: The abstract is poorly written without containing the key information regarding the findings of results obtained.
Thanks for the comment. As suggested, we revised the abstract by adding more findings in it. The revised version is shown below and the revised section is also highlighted in manuscript.
One type of low-cost and eco-friendly organic‒inorganic superabsorbent composite (SAPC) was synthesized by free radical polymerization of the acrylic acid (AA), starch (ST), sodium alginate (SA) and kaolin (KL) in aqueous solution. The structure and morphology of the SAPC were characterized by Fourier transform infrared spectrometer (FT-IR), scanning electron microscope (SEM), X-ray diffraction (XRD) and thermogravimetric analysis (TGA). The influence of different reaction conditions on the water absorption of SAPC, i.e., SA and KL contents, AA neutralization degree (ND), potassium persulfate (KPS) and N, N′-methylenebisacrylamide (MBA) loading were systematically studied. Under the optimal synthesis conditions, very high water absorption of 1200 g/g was achieved. The swelling kinetic mechanism of SAPC was studied by pseudo-second order swelling kinetics model and Ritger‒Peppas model. The performances of SAPC under different environments were tested and results revealed that this new SAPC has excellent swelling capacity, high water retention, good salt tolerance in monovalent salt solution (NaCl solution), and good pH tolerance between 4 and 10.
Comment 2: In the swelling test, the weight of the sample was constant (0.1 g). How was the weight controlled? I recommend recording the range.
The weight of the sample is within the weight range of 0.1 ± 0.001g. We made necessary modifications in the manuscript.
Comment 3: The results are presented in Figure 5 should discuss in detail based on the results obtained by depending on comparative values and differences of swelling tests.
Thanks for the comment. As suggested, we did the revisions as shown below:
- Page 7 line 25:
The water absorption increased from 736 to 933 g/g with the SA content changing from 10 wt. % to 15 wt. % and then decreased to 763 g/g with SA content further increased to 20 wt. %.
- Page 8 line 9:
Water absorption of the SAPC increased first and then decreased with the increase of KL content from 3 wt. % to 7 wt. %.
- Page 8 line 25.
The water absorption of the SAPC increased as the ND increased from 65% to 80%.
- Page 8 line 46.
Water absorption of the SAPC increased first and then decreased with the increase of MBA content from 0.1 wt. % to 0.2 wt. % and a maximum value of 1156 g/g was achieved with 0.15 wt. % MBA loading.
Comment 4: The conclusions obtained represent nothing in relation to the results obtained and should be supported by data.
Thanks for the comment. To respond the reviewer’s concern, we re-write the conclusions as follows:
To summarize, a series of SAPC was successfully prepared by free radical polymerization of SA, ST, AA and KL in aqueous solution. The optimum reaction conditions of SAPC were achieved when the mass of ST, SA, KL, KPS, and MBA were 15 wt. %, 15 wt. %, 4 wt. %, 0.9 wt. %, and 0.15 wt. % of AA, with neutralization degree of AA 80%. With optimized reactants concentrations, the maximum water absorption of 1200 g/g was obtained. FT-IR spectra confirmed the success of the polymerization reaction. SEM image revealed the rough and porous surface of SAPC which was conducive to the liquid entering into the polymer network. XRD patterns proved that KL was uniformly dispersed in the polymer matrix, and TGA spectra indicated that the addition of KL improved the thermal stability of SAPC. The swelling kinetic of SAPC was studied showing that the swelling behaviour of SAPC followed the pseudo-second order kinetic model and the non-Fick diffusion model. Except the excellent water absorption capacity, the fabricated SAPC also has excellent water retention property, good salt tolerance in monovalent salt solution and pH tolerance between 4 and 10, making it promising to be used in many applications.
